

# Spectral solutions for the Schrödinger equation with a regular singularity

Pushkar Mohile[1], Ayaz Ahmed[2], T.R.Vishnu[3] and Pichai Ramadevi[2*]

**1** Department of Physics, Stonybrook, USA
**2** Department of Physics, Indian Institute of Technology Bombay, India
**3** Department of Physics, Raman Research Institute, Bangalore, India

⋆ ramadevi@phy.iitb.ac.in

## Abstract

We propose a modification in the Bethe-like ansatz to reproduce the hydrogen atom spectrum and the wave functions. Such a proposal provided a clue to attempt the exact quantization condition (EQC) for the quantum periods associated with potentials $V(x)$ which are of the form $V(x) = |x| + a/|x| + b/|x|^2$. We validate the EQC proposal by showing that our computed Voros spectrum in the limit $a, b \to 0$ is matching well with the true spectrum of the familiar $|x|$ potential. Thus we have given a route to obtain the spectral solution for the one dimensional Schrödinger equation involving potentials with regular singularity at the origin.

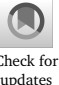

# 1  Introduction

The Bethe ansatz is one of the most powerful tools in the study of quantum integrable systems. It has profound applications in integrable spin chains. In fact, such an ansatz enables diagonalisation of Hamiltonian and obtains energy spectrum using simple algebraic arguments [1,2].

Even though the Bethe ansatz has been studied extensively in the context of integrable systems, there are interesting features that can be applied to other models. One such striking feature is to compute the quantum spectrum from the classical limit of the integrable model [3]. Such calculations are based on the asymptotes of a set of polynomial equations, which we get from the ansatz, called the Bethe equations. A nice review with applications to integrable quantum field theories and spin systems can be found in [4]. In fact, this review article and [3] illustrate the exact energy spectrum of one-dimensional quantum harmonic oscillator (QHO) from a Bethe-like ansatz. We believe such a neat concise Bethe-like ansatz approach must be generalisable for other quantum mechanical systems. Hence, we investigated this approach for the hydrogen atom and proposed a modification in the Bethe-like ansatz. Interestingly, we succeeded in reproducing the energy spectrum and the corresponding radial wave functions at all angular momenta.

Well behaved nature of wave function in quantum mechanics forces that the Bethe-like ansatz for pseudo-momentum $p_(x)$ (whose asymptotic behaviour in $\hbar \to 0$ matches classical momentum $p_{cl}(x)$) must have only simple poles. Unfortunately, we obtain higher order poles for any general polynomial potential of degree greater than 2. Hence, the Bethe-like approach fails for such potentials.

The natural extension of the Bethe-Like ansatz should be the exact WKB (Wentzel-Kramers-Brillouin) method [5,6] and the 'thermodynamic Bethe Ansatz' (TBA) equations governing the quantum WKB periods [7]. In fact, this route led to the energy spectrum for monic potentials [7] and general polynomial potentials [8]. TBA involves Borel transform and Borel re-summation techniques [6,8] to handle diverging series as well as capture the singularities in the Borel plane. In fact, these discontinuities in the Borel transform encode information about other perturbative series associated with different classical configurations [9,10].

For QHO and hydrogen atom, no new information is obtained using TBA approach. However, for higher order polynomial potentials, we can capture the information about the other zeros of the potential from the singularity structure of the Borel re-summed function. TBA approach is definitely powerful in computing the quantum periods for general polynomial potentials $V(x) = \sum_n a_n x^n$ [8]. Further, the exact WKB method advocated by Voros-Silverstone leads to an exact quantization condition (EQC) obeyed by the quantum periods [8,11,12]. Thus the spectral solutions for any general polynomial potentials can be obtained. They have been numerically presented for the polynomial potential with a suitable choice of parameters $\{a_n\}$ [8,11].

For other potentials with a simple pole and a double pole at the origin, the modification in the TBA analysis to obtain the quantum WKB periods has been systematically elaborated in [13]. However, due to the singularity at the origin, the EQC is still an open problem. The Bohr-Sommerfeld quantization to obtain the spectral solution is not correct.

The main theme of this paper is to propose a correction to the quantum period near the singular origin to modify the existing polynomial potential EQC. Such a proposal is motivated from our Bethe-like ansatz for the hydrogen atom. We validate our EQC proposal through an example whose energy spectrum is known.

We know that the wave functions for the $|x|$ potential are the Airy functions. In fact, the zeros of the Airy function and its derivatives give the true spectrum [14, 15]. We performed a naive TBA approach for the $|x|$ potential with two turning points and obtained the spectrum using the EQC of the QHO. Our calculated spectrum did not match the true spectrum for the low lying energy states. This exercise indicates that the conventional (TBA & EQC) approach, of finding spectral solutions for polynomial potentials, cannot be applied to the potentials with a derivative singularity at the origin.

Incidentally, the solutions for $|x|^{2n+1}$ potentials using spectral determinant approach are discussed in [5, 16]. However, our aim was to investigate the Voros-Silverstone connection formulae and propose a EQC to reproduce the true spectrum for these potentials.

As a first step, we showed that the $|x|$ potential can be viewed as the potential with a simple and a double pole [13] for a suitable limit of parameters. With this choice of parameters, we numerically computed the quantum periods using TBA equation. Then, using our EQC proposal we obtained the Voros spectrum. In fact, our numerical results for the Voros spectrum match well with the true spectrum. Our validation for $|x|$ spectrum reinforces that the proposed EQC is applicable for the potentials with a regular singularity [13].

The plan of the paper is as follows: In section 2, we briefly review Bethe-like ansatz and present the spectrum of QHO. Then we propose a modification in the Bethe-like ansatz necessary to reproduce the hydrogen atom spectrum. In section 3, we have discussed exact WKB method and Borel resummation technique to deal with divergent perturbative series. We summarise the salient details of TBA in section 4 with some simple potentials as illustrative examples. We discuss the $|x|$ potential and its relation to potentials with regular singularity in 4.2. In section 5, we focus on our proposals of EQC for potentials, singular at the origin. We summarise and present some of the open problems in the concluding section 6.

## 2 Bethe-Like Ansatz

For completeness and clarity, we will first review the salient features of Bethe-like ansatz approach for the quantum harmonic oscillator (QHO) spectrum [3, 4]. Then, we present our proposal of modified Bethe-ansatz giving hydrogen atom spectrum.

For QHO, a set of Bethe-like equations can be written for the roots of the wave function. This relies on the nonlinear transformation of time independent Schrödinger equation (TISE)

$$-\frac{\hbar^2}{2m}\frac{d^2}{dx^2}\psi(x) + V(x)\psi(x) = E\psi(x),\tag{1}$$

into Riccati equation:

$$p^2 - i\hbar p' = 2m(E - V),\tag{2}$$

where

$$p(x) = \frac{\hbar}{i}\frac{\psi'(x)}{\psi(x)}.\tag{3}$$

Note that $p(x)$(3) has singularities at the zeros of the wave function $\psi(x)$. Such singularities are handled by doing an analytic continuation of $p(x) \to p(z)$ in the complex plane. The nature of *complex function $p(z)$* can be fixed from the generic behaviour of the wave function $\psi(z)$, i.e., $\psi(z)$ must be normalisable. Suppose we allow second or higher order poles for $p(z)$,

$$p(z) \propto (z - a)^{-n} \text{ for } n \geq 2.\tag{4}$$

Then the wave function

$$\psi(z) \propto \exp\left(\frac{i}{\hbar} \int p(z)dz\right) = \exp[-\frac{i}{\hbar}\frac{(z-a)^{-n+1}}{n-1}], \tag{5}$$

has essential singularities. This implies that $p(z)$ can have at most simple poles. Note that the roots of the bound state wave functions $\psi(z)$ are discrete and isolated [17].

For highly excited states, we can take the classical limit $\hbar \to 0$. Clearly, $p(z)$ in the classical limit

$$\lim_{\hbar \to 0} p(z) \equiv p_{\mathrm{cl}}(z) = \pm\sqrt{2m(E-V(z))}, \tag{6}$$

denotes the familiar classical momentum of the particle which has branch cut singularity. It must be puzzling as to where from this branch cut emerges in the classical limit. It can only be formed when the discrete poles present in $p(z)$ 'condense' to a continuum as we approach the classical limit. Hence, we can conclude that the poles condense into the branch cut in the classical limit. Note that $p(z)$ is applicable for classically allowed region ($E \geq V(z)$) as well as classically forbidden region ($E < V(z)$). Hence $p(z)$ is referred to as pseudo-momentum.

We will now review QHO spectrum from the Riccati equation to see the resemblance with Bethe ansatz equations.

Let us examine the classical limit $p_{\mathrm{cl}}(z)$(6) for QHO, of mass $m$ and angular frequency $\omega$ , whose $V(z) = V_{\mathrm{QHO}} = m\omega^2 z^2/2$. The function (6) has a square root type branch cut, with branch points at the two turning points

$$z = \pm\sqrt{2E/m\omega^2}.$$

Our aim is to determine the allowed energy eigenvalues $E$ for QHO. In order to achieve this, we probe the asymptotic behaviour of $p(z)$ as $z \to +\infty$ on the real axis :

$$p \sim im\omega z + \mathcal{O}(\frac{1}{z}) \ \text{ and } \ p' \sim im\omega + \mathcal{O}(\frac{1}{z^2}).$$

Notice that the leading term in asymptotic $p'(z)$ is a constant and must be included so that Riccati equation gives

$$\begin{aligned}
\lim_{z\to\infty} p(z) \equiv p_o(z) &= \lim_{z\to\infty} \sqrt{2m\left[(E-\frac{\hbar\omega}{2}) - \frac{m\omega^2 z^2}{2}\right]} \\
&\sim im\omega z - i\frac{(E-\hbar\omega/2)}{\omega z} + \mathcal{O}(1/z^3),
\end{aligned} \tag{7}$$

where $\lim_{\hbar\to 0} p_o(z) = p_{\mathrm{cl}}(z)$(6). Notice that the asymptotic behaviour of $p_o(z)$ is almost like the classical momentum(6), if we shift

$$E \to E - \frac{1}{2}\hbar\omega. \tag{8}$$

In fact, the branch cut of $p_o(z)$ includes all the poles of $p(z)$. It is exciting to see the natural emergence of quantum shift in the energy by $\hbar\omega/2$ from the Riccati equation for QHO. The asymptotic behaviour of $p(z) \equiv p_o(z)$, which has a branch cut, is due to the condensation of simple poles of $p(z)$. This leads to the following Bethe-like ansatz for $p(z)$ having $N$-simple poles:

$$p(z) = im\omega z + \frac{\hbar}{i}\sum_j^N \frac{1}{z-z_j}. \tag{9}$$

Here, we make the choice of sign in the leading term ($im\omega z$) so that the wave function remains normalisable. The set $\{z_j\}$ corresponds to the $N$ roots arising from the nodes of the

Table 1: Bethe equations for QHO roots for $\frac{\hbar}{m\omega} = 1$.

| $N$ | Equations | Solutions |
|---|---|---|
| 0 | No equations | No roots |
| 1 | $z_1 = 0$ | $z_1 = 0$ |
| 2 | $z_1(z_1 - z_2) = 1$ | $z_1 = -1/\sqrt{2}$ |
|   | $z_2(z_2 - z_1) = 1$ | $z_2 = 1/\sqrt{2}$ |
| 3 | $z_1(z_1 - z_2)(z_1 - z_3) = 2z_1 - z_2 - z_3$ | $z_1 = -\sqrt{3/2}$ |
|   | $z_2(z_2 - z_1)(z_2 - z_3) = 2z_2 - z_1 - z_3$ | $z_2 = 0$ |
|   | $z_3(z_3 - z_1)(z_3 - z_2) = 2z_3 - z_1 - z_2$ | $z_3 = \sqrt{3/2}$ |

$N^{th}$ excited eigenfunction. Incorporating the key observation of the Bethe-like approach, the contour integration around the branch cut in $p_o(z)$ must give the residues due to the simple poles of $p(z)$(9):

$$\oint_\gamma \sqrt{2m\left[\left(E - \frac{\hbar\omega}{2}\right) - V(z)\right]}dz = \sum_j^N \text{Res}_{z_j} = 2\pi\hbar N, \tag{10}$$

where $N$ is the number of roots for the $N^{th}$ excited state and $\gamma$ is a contour around the branch cut. By doing this contour integral, we get

$$2\pi\left(E - \frac{\hbar\omega}{2}\right) = 2\pi(N)\hbar\omega, \tag{11}$$

which gives us the energy spectrum of the QHO:

$$E = \left(N + \frac{1}{2}\right)\hbar\omega. \tag{12}$$

We have to deduce the wave functions $\psi_N(z)$ corresponding to the $N$-th excited energy level from the Riccati equation. When we substitute the ansatz(9) into the Riccati equation(2) and equate the coefficients of each of the terms $1/(z - z_j)$ to zero, we obtain Bethe-like equations for the the roots $\{z_j, j = 0, 1, \ldots N\}$ of the $N$-th excited state wavefunction $\psi_N(z)$ :

$$z_j = \frac{\hbar}{m\omega}\sum_{i \neq j}\frac{1}{z_j - z_i}, \quad \forall j = 1, 2, 3, ..N. \tag{13}$$

This system of polynomial equations is solvable, with solutions to $z_j$ being the roots of Hermite polynomials when the factor $\hbar/m\omega$ is scaled to 1. We have solved this system of polynomial equations for $N \leq 3$ using Mathematica and tabulated (see Table 1). These roots computed are matching with the roots of Hermite polynomials. Once we know the roots $\{z_j\}$'s for any $N$, the corresponding energy eigenfunction is constructed as

$$\psi_N(z) = \exp\left[\frac{i}{\hbar}\int p(z)dz\right] = A\exp\left(\frac{-iz^2}{2\hbar}\right)\prod_{j=1}^N(z - z_j), \tag{14}$$

where $A$ is determined by normalisation. The flowchart (Table 2) gives a concise summary of the Bethe-like methodology for QHO. Thus we have elaborated the powerfulness of this Bethe-like approach to obtain the complete QHO energy spectrum and the corresponding wave functions. Particularly, the branch cut in asymptotic behaviour of $p_o(z)$ is accountable by condensation of simple poles. Further, the well known zero point energy ($\hbar\omega/2$) appeared naturally. It is not clear whether this methodology works for arbitrary potential $V$. As a first step in this direction, we have attempted hydrogen atom in the following subsection.

Table 2: Flowchart for QHO spectrum from Bethe-Like approach.

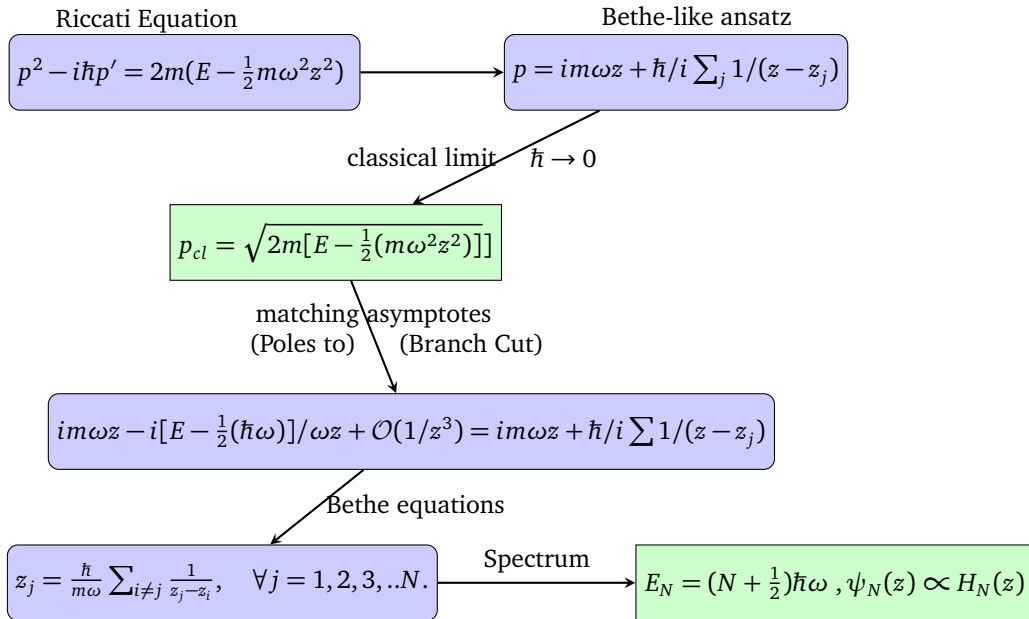

## 2.1 Bethe-Like Ansatz for the Hydrogen Atom

For the hydrogen atom, the potential energy is $V(r) \propto \frac{1}{r}$. Clearly, rotations in the three-dimensional space leaves the Hamiltonian of the hydrogen atom invariant. Even though it is a three-dimensional system, we can view the hydrogen atom as an effective one dimensional system in radial coordinate $r$. By rewriting the radial part $R_n^l(r)$ of the wavefunction $\psi_{n,l,m}(r,\theta,\phi) = R_n^l(r)Y_{lm}(\theta,\phi)$ as $u_n^l(r) = rR_n^l(r)$, it is easy to check that the radial part of the equation resembles the one-dimensional Schrödinger equation for $u_n^l(r)$ with effective potential energy

$$V_{eff}(r) = -\frac{e^2}{4\pi\epsilon_o r} + \frac{\hbar^2 l(l+1)}{2mr^2}, \tag{15}$$

where the quantum number $l$ refers to the orbital angular momentum. Following the arguments in the previous section, the pseudo-momentum $p(r)$ is a rational function with simple poles at $\{r_j\}$ for regular functions $u_n^l(r)$ having zeros at the points $\{r_j\}$ where $j = 1, 2, \dots N$. The residues of $p(r)$ at these poles must be $2\pi i \times \hbar/i$. We should keep in mind the following key differences between the harmonic oscillator and the hydrogen atom:

(i) The passage from the three-dimensional problem to the effective one-dimensional system should introduce an additional pole at $r = 0$.

(ii) the domain of definition is $r \geq 0$.

(iii) The asymptotic form of pseudo-momentum $p(r)$ matches exactly the asymptotic values of the classical momentum $p_{cl}(r)$.

Hence, for highly excited states, $p(r)$ is the classical momentum (there is no zero point energy shift). In the classical limit, we observe the poles at $\{r_j\}$ condense to form a square root branch cut of $p(r)$. At large values of $r$, $p \to -\sqrt{2mE}$ as $V \to 0$, where the negative branch of the square root is chosen to prevent the wave function $u(r)$ from blowing up at $\infty$. We will now focus on the energy and the wave function for the s-orbitals of hydrogen atom ($l = 0$) which will give us the insight to generalise the Bethe-like ansatz for $l \neq 0$.

### 2.1.1 Spectrum for $l = 0$

Let us propose an ansatz for $p(r)$ for the $s$-orbitals whose orbital angular momentum $l = 0$ to obtain the energy spectrum and the corresponding wave function $u(r)$. Incorporating the asymptotic form of $p(r)$ and its poles at $\{r_j\}$, we propose the following Bethe-like ansatz.
**Proposal 1:**

$$p = -\sqrt{2mE} + \frac{\hbar}{i}\frac{1}{r} + \frac{\hbar}{i}\sum_{j=1}^{N}\frac{1}{r - r_j}. \tag{16}$$

with $N + 1$ poles including the pole at $r = 0$. Recall this additional pole was not there in the harmonic oscillator. The large $r$ limit can be expressed as

$$\lim_{r\to\infty} p(r) = \lim_{r\to\infty} -\sqrt{2mE\left(1 - \frac{b}{rE}\right)} \sim -\sqrt{2mE} + \frac{b\sqrt{m}}{r\sqrt{2E}} + \mathcal{O}\left(\frac{1}{r^2}\right), \tag{17}$$

where $b = -e^2/4\pi\epsilon_0$. Since the poles must condense to this branch cut, on doing a contour integration around the set of zeroes $\{r_j\}$, the residues must equate on both sides giving us

$$\frac{b\sqrt{m}}{\sqrt{2E}} = \frac{\hbar}{i}(N + 1). \tag{18}$$

Here, $N$ is the number of zeros of the $s$ orbital wavefunction $u_n^{l=0}(r)$, and one more pole from $r = 0$ for $R_n^0(r)$. On rearranging and substituting $b = -e^2/4\pi\epsilon_0$ we get

$$E_n = -\frac{e^4 m}{32\hbar^2\pi^2\epsilon_0^2 n^2} = -\frac{e^2}{2a_0}\frac{1}{n^2} \text{ where } n = N + 1, \tag{19}$$

where $a_0 = \hbar^2/(me^2)$ is the Bohr radius (set $4\pi\epsilon_0 = 1$). This matches exactly with the hydrogen atom energy spectrum. Further, (16) allows us to fix the roots of the wave function $u(r)$ by requiring that the coefficients of each of the $1/(r - r_i)$ terms add up to zero in the Riccati equation:

$$\sqrt{2mE_n} = \frac{\hbar}{i}\frac{1}{r_i} + \frac{\hbar}{i}\sum_{\{k\neq i\}=1}^{N}\frac{1}{r_i - r_k} \quad \forall i \in 1, 2, ..N. \tag{20}$$

This gives us a set of Bethe-like equations to solve and determine the roots $\{r_i\}$. For $N = 1$, we get $r_1 = 2a_0$. We have tried to work out the roots for $N = n - 1 \leq 3$ using Mathematica and presented the results in Table 3 for $a_0 = 1$. Once we have explicitly found the roots, we can then integrate the ansatz for $p$ to obtain the wave function $R_n^{l=0}(r)$. We see explicitly that for $N$ zeros of the wave function $u(r)$ we get

$$R_n^0(r) = A\exp(-\sqrt{2m|E_n|}\, r)L_n^0(r). \tag{21}$$

where $L_n^0(r)$ are the Laguerre polynomials. Here the negative branch of the square root is chosen to ensure $R_n(r) \to 0$ as $r \to \infty$ and $A$ is the normalisation constant. $L_n^0(r) = \prod_{i=1}^{N}(r - r_i)$ is a polynomial with roots at $r_i$ found from Bethe-like equations. For $N = 0$ and $N = 1$, we get the wave functions

$$R_1^0(r) = A_1\exp(-\frac{r}{a_0}), \quad R_2^0(r) = A_2\exp(-\frac{r}{2a_0})(r - 2a_0). \tag{22}$$

Using the mathematica program, we can deduce $N = 2$ for $a_0 = 1$ as

$$R_3^0(r) = A_3\exp(-\frac{r}{3})(r - 3/2(3 - \sqrt{3}))(r - 3/2(3 + \sqrt{3})). \tag{23}$$

We will generalise the ansatz for $p(r)$ for arbitrary $l$ in the following subsection.

Table 3: Bethe equations for Hydrogen atom roots for $l = 0$ and $a_0 = 1$.

| $n = N + 1$ | Equations | Solutions |
|---|---|---|
| 1 | No equations | No roots |
| 2 | $r_1 = 2$ | $r_1 = 2$ |
| 3 | $r_1^2 - r_1 r_2 - 6r_1 + 3x_2 = 0$<br>$r_2^2 - r_1 r_2 - 6r_2 + 3r_1 = 0$<br>$1/r_1 + 1/r_2 = 2/3$ | $r_1 = 3/2(3 - \sqrt{3})$<br>$r_2 = 3/2(3 + \sqrt{3})$ |
| 4 | $r_1(r_1 - r_2)(r_1 - r_3) = 4(3r_1^2 - 2r_1 r_2 - 2r_1 r_3 + r_2 r_3)$<br>$r_2(r_2 - r_1)(r_2 - r_3) = 4(3r_2^2 - 2r_2 r_1 - 2r_2 r_3 + r_1 r_3)$<br>$r_3(r_3 - r_1)(r_3 - r_2) = 4(3r_3^2 - 2r_3 r_1 - 2r_3 r_2 + r_1 r_2)$<br>$1/r_1 + 1/r_2 + 1/r_3 = 3/4$ | $r_1 = 1.871$<br>$r_2 = 6.618$<br>$r_3 = 15.517$ |

### 2.1.2 Spectrum for $l \neq 0$

**Proposal 2:** Generalising *proposal 1* in (16), for any $l$, Bethe-like ansatz for $p(r)$ is

$$p = -\sqrt{2mE} + \frac{\hbar}{i} \left[ \frac{l+1}{r} + \sum_j \frac{1}{r - r_j} \right]. \tag{24}$$

Such an ansatz will take care of the additional multiplicity of the root at $r = 0$ for wave function $R_n^l(r)$. Further, the term $l(l+1)/r^2$ term in the effective potential will be accounted for by the modified ansatz.

The calculation of the energy spectrum $E_n$ for the modified ansatz is almost the same:

$$E_n = -\frac{e^4 m}{32 \hbar^2 \pi^2 \epsilon_0^2 n^2} = -\frac{e^2}{2a_0} \frac{1}{n^2} \text{ where } n = N + l + 1. \tag{25}$$

Note that $n$ counts the total number of roots of the wave function $R_n^l(r)$ and $l + 1$ counts the degeneracy of the root at the origin $r = 0$. Interestingly, we observe the bound on $l$ to be:

$$l \leq n. \tag{26}$$

Substituting the modified ansatz in the Riccati equation and equating the coefficients of $1/(r_i - r_j)$ to zero we get the following set of Bethe equations for the roots:

$$\sum_j \frac{1}{r_j} = \frac{1}{a_0} \left( \frac{1}{l+1} - \frac{1}{N+l+1} \right), \tag{27}$$

$$\frac{\hbar}{i} \sqrt{2mE_n} = \hbar^2 \sum_{i \neq j} \frac{1}{r_j - r_i} + \hbar^2 \frac{(l+1)}{r_j}, \ j = 1, 2, ..N. \tag{28}$$

Solving these equations for every $N$ will give the solutions for the roots leading us to write the associated Laguerre polynomials:

$$R_n^l(r) \propto L_n^l(r). \tag{29}$$

From our **proposal 1 and 2** of reproducing hydrogen atom spectrum, it is tempting to speculate whether the spectrum for arbitrary potential $V(x) = \sum_k a_k x^{\pm k}$ can be elegantly obtained. Unfortunately, well behaved nature of wave function requiring $p(z)$ to have only simple poles is inconsistent with the asymptotic expansion of classical momentum $p_{cl}(z)$:

$$\lim_{z \to \infty} \sqrt{2m(E - z^n)} \sim i\sqrt{2mz^n} \left( 1 - \frac{E}{z^n} + \mathcal{O}\left( \frac{1}{z^{2n}} \right) \right), \quad \text{for} \quad n > 0, \tag{30}$$

$$\lim_{z \to \infty} \sqrt{2m(E - z^n)} \sim \sqrt{(2mE)}(1 - z^n/E + \mathcal{O}(z^{2n})), \quad \text{for} \quad n < 0. \tag{31}$$

It appears that the Bethe-like ansatz requires the wave function to factorise into two parts:

$$\psi(x) = f(x)g(x) \; ; \; p(x) = \frac{\hbar}{i}(f'/f + g'/g). \tag{32}$$

Here $f(x)$ governs the asymptotic behaviour of $\psi(x)$ in the limit $|x| \to \infty$ and $g(x)$ is the polynomial that encodes the roots of $\psi(x)$. In the Bethe-like ansatz, we assumed that the asymptotic behaviour of the wave function is governed only by the leading order asymptotic behavior, which looks like $\exp[-x^2]$ for the QHO and $\exp[-r/na_0]$ for the hydrogen atom. This is the most trivial possible choice of the asymptotic behaviour of the function. Such a choice of asymptote does not appear for other potentials. We may have more contributions to the asymptote due to non-perturbative corrections. Hence, we will have to go beyond Bethe-like ansatz to tackle spectral solution for higher degree polynomial potentials.

# 3 WKB Method

The key features of the Bethe-like ansatz were the computation of the period of the pseudo-momentum around the classically allowed region and matching this period with the counting of poles to get the quantisation condition. For a general polynomial potentials, there will be many periods with some around the classically forbidden regions giving exponentially suppressed tunnelling contribution. The conventional WKB (Wentzel-Kramers-Brillouin) approximation involves matching of region dependent wave function at the interface giving WKB quantisation. However, the energy spectrum matches only for large quantum numbers (semi-classical limit). Going beyond the conventional WKB to 'Exact WKB method' requires the tools of Borel transform, Borel resummed WKB periods. Then the connection formulae, involving such Borel resummed WKB periods, advocated by Voros-Silverstone will lead to the 'Exact quantization condition (EQC)' giving the true energy spectrum.

In this section, we briefly review the concepts of Borel resummation required for the Exact WKB. The computation of the exact Borel resummed WKB periods, using 'Thermodynamic Bethe Ansatz (TBA)' equations, will be discussed in the subsequent sections for polynomial potentials as well as the potentials with a regular singularity.

## 3.1 Exact WKB Theory

In order to find perturbative corrections to the spectrum, we expand the pseudo-momentum $p(x)$ as power series in $\hbar$

$$p(x) = \sum_{n=0}^{\infty} p_n(x)\hbar^n, \tag{33}$$

with $p_o = p_{cl} = \pm\sqrt{2m(E - V(x))}$. The solutions $p_n(x)$ can be computed order by order using the Ricatti equation. In order to compute the spectrum, we need to compute the periods of this solution :

$$\Pi_\gamma(\hbar) = \oint_\gamma p(x)dx = \oint_\gamma \sum_n p_n(x)\hbar^n dx = \sum_{n=0}^{\infty} \Pi_{\gamma,n}\hbar^n, \tag{34}$$

where the integral is around the classically allowed region $\gamma$.

However, it is well known that the perturbative series obtained diverges. Explicit numerical analysis for example shows that the quantum periods for monic potentials $x^{2m}, m > 1$ diverge factorially [18]. The resolution of this divergence can be understood through the exact WKB theory developed by Voros and others [6,8] .

In any quantum system, the space of classical configurations is given by the extrema of the potential $V(x)$ (also called saddle points). QHO has one extremum whereas the cubic

potential $V(x) = 3x^2 - x^3$ illustrated in Figure 1 has two extrema. Technically, we have to investigate perturbative series around each of these classical configurations. They will give different quantum periods $\Pi_{\gamma_1}, \Pi_{\gamma_2}, \ldots$ For instance, there are two periods in Figure 1 corresponding to the curves $\gamma_1$(classically allowed region) and $\gamma_2$(classically forbidden region). Hence, the divergent series $\Pi_\gamma$ (34) implicitly signals the presence of other perturbative series in the quantum system. This is the theme of resurgent quantum mechanics [9]. In order to capture such information, we need the tools of 'Exact WKB methods' advocated by Voros for higher degree polynomials We recall the key notions of Borel resummation and EQCs we will need for this approach.

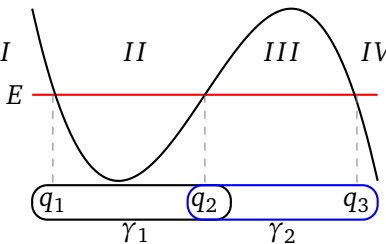

Figure 1: WKB loops for potential $V = 3x^2 - x^3$.

In the series solution to the Ricatti equation 33, it is known that that the terms involving odd powers are dependent on the terms containing even powers. Thus we rewrite the ansatz only in terms of the even powers $P(x) = \sum_{n=0}^{\infty} p_{2n}(x)\hbar^{2n}$. The general wavefunction in this case is given by :

$$\psi(x) = \frac{1}{\sqrt{P(x)}} \left( A e^{i \int P(x)dx} + B e^{-i \int P(x)dx} \right),\tag{35}$$

where $A, B$ are normalisation constants . We need to change $P(x)$ to $\tilde{P}(x) = iP(x)$ when we move to the classically forbidden region.

In the exact-WKB approach, the divergent power series will be converted to a series with finite radius of convergence by a Borel transform. For the perturbative series discussed for monic potentials, Borel transform is as follows [8]:

$$\Pi_\gamma(\hbar) \to \hat{\Pi}_\gamma(\xi) = \sum_n \hat{\Pi}_{\gamma,2n} \xi^{2n} = \sum_n \frac{\Pi_{\gamma,2n}}{2n!} \xi^{2n},\tag{36}$$

which is analytic near origin in the complex plane $\xi$. Then, the $\hat{\Pi}_\gamma(\xi)$ is promoted to a function through a procedure called 'Borel resummation' :

$$\mathcal{B}_\phi[\Pi_\gamma](\hbar) = \frac{1}{\hbar} \int_0^{e^{i\phi}\infty} e^{-\xi/\hbar} \hat{\Pi}_\gamma(\xi)d\xi , \quad \hbar \in \mathbb{R}_{>0}.\tag{37}$$

The above integral denotes a Laplace transform of $\hat{\Pi}_\gamma(\xi)$ along a direction, defined by angle $\phi$, in the complex plane $\xi$. Here, the upper limit of the integral $e^{i\phi}\infty$ means integrating from the origin to infinity along this ray. If the integral converges for small $\hbar$, then the corresponding quantum period $\Pi_\gamma(\hbar)$ is said to be Borel summable. Suppose $\hat{\Pi}_\gamma(\xi)$ has singularities on the

complex plane $\xi$, then the Borel summability cannot be performed on the rays containing such singularities. For instance, a simple pole at $\xi_0$ whose $\arg \xi_0 = \chi$ will imply a discontinuity in the Borel resummation. That is.,

$$\lim_{\delta \to 0} \mathcal{B}_{\chi+\delta}[\Pi_\gamma](\hbar) \neq \lim_{\delta \to 0} \mathcal{B}_{\chi-\delta}[\Pi_\gamma](\hbar).$$

Hence, we define median Borel resummation $\mathcal{B}_\chi^{med}[\Pi_\gamma](\hbar)$, the lateral Borel resummation $\mathcal{B}_{\chi\pm}[\Pi_\gamma](\hbar)$ and the Stokes discontinuity $\mathrm{disc}_\chi[\Pi_\gamma](\hbar)$ to characterise and overcome such obstructions to Borel summability:

$$
\begin{aligned}
\mathcal{B}_\chi^{med}[\Pi_\gamma](\hbar) &= \frac{1}{2} \lim_{\delta \to 0} (\mathcal{B}_{\chi+\delta}[\Pi_\gamma](\hbar) + \mathcal{B}_{\chi-\delta}[\Pi_\gamma](\hbar)), &(38)\\
B_{\chi\pm}[\Pi_\gamma](\hbar) &= \lim_{\delta \to 0} B_{\chi\pm\delta}[\Pi_\gamma](\hbar), &\\
\mathrm{disc}_\chi[\Pi_\gamma](\hbar) &= \lim_{\delta \to 0} (B_{\chi+\delta}[\Pi_\gamma](\hbar) - B_{\chi-\delta}[\Pi_\gamma](\hbar)). &
\end{aligned}
$$

The knowledge of all the Stokes discontinuities as well as the classical limit $\Pi_{\gamma,0}$ of the quantum periods are required to reconstruct the quantum periods (as solutions to the Riemann-Hilbert problem). The *Delabaere-Pham formula* [9,10] encodes the structure of discontinuities of any quantum period in terms of the other quantum periods:

$$\mathcal{B}_{\chi-}(\mathcal{V}_{\gamma_i}) = \mathcal{B}_{\chi+}(\mathcal{V}_{\gamma_i}) \prod_{j \neq i} (1 + \mathcal{V}_{\gamma_j}^{-1})^{-(\gamma_i, \gamma_j)}. \tag{39}$$

Here,

$$\mathcal{V}_{\gamma_i} = \exp\left(\frac{i \Pi_{\gamma_i}}{\hbar}\right)$$

is called as *Voros symbol* and $(\gamma_i, \gamma_j)$ is the intersection number between the curves $\gamma_i, \gamma_j$. Once we have the solutions for all the Voros symbols, the *exact WKB connection formula* (also known as Voros-Silverstone connection formulae) leads to an *exact quantization condition* (EQC) as a single functional relation between $\mathcal{V}_{\gamma_i}$'s:

$$f(\mathcal{V}_{\gamma_1}, \mathcal{V}_{\gamma_2}, \dots) = 0. \tag{40}$$

For example, using the Voros-Silverstone connection formulae for the cubic potential $V(x) = 3x^2 - x^3$, the following EQC relating the two quantum periods (as drawn in Figure 1) can be deduced [8,11]:

$$2 \cos\left(\frac{1}{2\hbar} \mathcal{B}_{\chi\pm}(\Pi_{\gamma_1})\right) + \exp\left(-\frac{i}{\hbar} \Pi_{\gamma_2}\right) = 0, \tag{41}$$

Recall that the $\Pi_{\gamma_2}$ is associated with the classically forbidden region. Such a relation gives the values of energy $E_n$. Sometimes it is convenient to fix the value of energy and compute the values of $\hbar_n(E)$ for which the EQC(40) holds. These values of $\hbar_n(E)$ are called *Voros spectrum*.

As mentioned earlier, the solution to the Riemann-Hilbert problem (quantum periods) can be obtained from a set of '*Thermodynamic Bethe Ansatz*'(TBA) equations. We will briefly discuss TBA method in the following section.

## 4 TBA system

We will first briefly review TBA integral equations for polynomial potentials. Then we highlight the shortfall of using the approach for other potentials like $|x|$ with derivative singularity at the origin. Interestingly, a modification of the TBA equations for potentials with single and double poles [13] suggests how to infer the quantum periods for $|x|$. We will present these details in this section.

## 4.1 Polynomial potentials

For monic potentials, including quartic oscillator, $V(x) = x^{2M}$, there are $2M$ turning points located at $\{w^i E^{\frac{1}{2M}}\}$ in the complex plane where $\omega$ is $2M$-th root of unity and $E$ is the energy. Only two of the turning points are on the real axis. Similarly, for a general $d$-degree polynomial potentials $V(x) = \sum_{n=1}^{d} a_n x^n$, there will $d$ turning points. Depending on the choice of $a_n$ (known as moduli), the turning points could be real or complex. We can make a suitable choice of the moduli so that all the turning points $x_1 < x_2 < \ldots < x_d$ are on the real axis. This choice is sometimes referred to as '*minimal chamber*' in the literature. Such a minimal chamber will allow $\lfloor (d-1)/2 \rfloor$ cycles $\{\gamma_i\}$. In fact, the cubic potential $V(x) = 3x^2 - x^3$ shown in Figure 1 allows two cycles $\gamma_1, \gamma_2$ in the minimal chamber.

In such a minimal chamber, the quantum periods $\Pi_{\gamma_{2i}}$ corresponding to classically forbidden region are Borel summable along the positive real axis of $\hbar$ whereas $\Pi_{\gamma_{2i-1}}$ corresponding to classically allowed region are not Borel summable. Hence the discontinuity formula(39) along the real line ($\chi = 0$) is

$$\mathrm{disc}_0 \Pi_{2i-1} = -i\hbar \log(1 + \mathcal{V}_{2i-2}^{-1}) - i\hbar \log(1 + \mathcal{V}_{2i}^{-1}),. \tag{42}$$

Similarly, there is a discontinuity at $\chi = \pi/2$ for the quantum periods $\Pi_{2i}$ whereas $\Pi_{\gamma_{2i-1}}$ are Borel summable. These two situations are neatly incorporated by defining $\epsilon_a$ functions as:

$$-i\epsilon_{2i-1}(\theta + i\pi/2 \pm i\delta) = \frac{1}{\hbar}\mathcal{B}_{0\pm}(\Pi_{\gamma_{2i-1}})(\hbar) \tag{43}$$

$$-i\epsilon_{2i}(\theta) = \frac{1}{\hbar}\mathcal{B}(\Pi_{\gamma_{2i}}), \tag{44}$$

where $e^\theta = 1/\hbar$ [8]. Clearly, these $\epsilon_a$ functions have a discontinuity at $\chi = \pi/2$ for both even and odd $a$. Hence, the Delabaere-Pham discontinuities(39) can be compactly written as:

$$\mathrm{disc}_{\pi/2}\epsilon_a(\theta) = L_{a-1}(\theta) + L_{a+1}(\theta) \quad ,\text{where} \quad L_a = \log(1 + e^{-\epsilon_a(\theta)}). \tag{45}$$

Further, the asymptotic series of the functions $\epsilon_a(\theta)$ will be

$$\epsilon_a(\theta) = m_a e^\theta + \mathcal{O}(e^{-\theta}), \tag{46}$$

where $m_a$'s, referred to as masses in two-dimensional integrable theories, are the classical periods:

$$m_a = \Pi_{\gamma_a, 0} = \oint_{\gamma_a} p_0(x)dx = 2\int_{x_a}^{x_{a+1}} p_0(x)dx \quad \text{where} \quad \gamma_a = [x_a, x_{a+1}]. \tag{47}$$

Remember to replace $P(x) \to iP(x)$ whenever the cycle $\gamma_{a \equiv 2i}$ ( classically forbidden region) so that $m_a$'s are real and positive.

The solution to the Riemann Hilbert problem for the functions $\epsilon_a(\theta)$ obeying (45) and (47) can be obtained using the following system of TBA integral equations in the minimal chamber:

$$\epsilon_a(\theta) = m_a e^\theta - \int_{\mathbb{R}} \frac{L_{a-1}(\theta')}{\cosh(\theta - \theta')}d\theta' - \int_{\mathbb{R}} \frac{L_{a+1}(\theta')}{\cosh(\theta - \theta')}d\theta' \qquad a = 1, 2, \ldots d-1, \tag{48}$$

As $P(x)$ is a series in even powers of $\hbar$, we have to take both $\hbar$ positive as well as negative. This in turn adds another similar discontinuity equation, and combining all of these discontinuities transforms the usual propagator into the sinh propagator. Finally, the rotation by $\pi/2$ gives

us the $\cosh(\theta - \theta')$ in the above integral equation[1]. For other choices of moduli$\{a_n\}$ in the potential $V(x) = \sum_n a_n x^n$, some turning points can be on the complex plane. This leads to additional periods in the complex plane. We do not review calculations involving complex turning points here. This is discussed in great detail in [11].

## 4.2 TBA for $|x|$

We have seen in the previous subsection how the TBA system can be used to compute quantum periods for the polynomial potentials which are smooth [7,8] and deduce the Voros spectrum from EQC. Can this TBA equations be blindly applied to other potentials with derivative singularity at the origin? In the literature, such potentials of the form $|x|^n$ with $n$ odd positive integer were considered in [5] using exact WKB method and spectral determinants but without TBA equations. Further, a TBA equation was derived in [19] albeit from very different considerations of 4D $\mathcal{N} = 2$ supersymmetric field theories. In fact, this was argued to be the spectral determinant of the $|x|$ potential which was expanded upon in [16]. These developments are suggestive of the existence of TBA equations for such potentials.

### 4.2.1 Naive TBA approach for $|x|$

Let us blindly apply the TBA tools, applicable for polynomial potentials, to deduce the WKB periods for the $|x|$ potential. The Schrödinger equation is given by

$$\hat{H}\psi = -\frac{d^2}{dx^2}\psi + |x|\psi = E\psi. \tag{49}$$

In this case, there are two turning points, corresponding to $x = +E$ and $x = -E$. As there is only one nontrivial cycle $\gamma$ between them as seen in Figure 2, the solution to the TBA equation (48) for this period is the mass $m$. As there is only one cycle, we believe that the EQC for $|x|$ will be similar to the QHO case:

$$\frac{\Pi_\gamma}{\hbar} = \frac{m}{\hbar} = 2\pi\left(n + \frac{1}{2}\right), \qquad n = 0, 1, 2\ldots \tag{50}$$

where the $0^{th}$ order mass $m$ is given by

$$m = \oint_\gamma \sqrt{E - |x|}\, dx = 2\int_{-E}^{E} \sqrt{E - |x|}\, dx = \frac{8E^{\frac{3}{2}}}{3} \tag{51}$$

On solving the Bohr-Sommerfeld quantisation condition(50) we obtain the spectrum for $\hbar = 1, 2m = 1$

$$E_n = \left(\frac{3\pi}{4}\left(n + \frac{1}{2}\right)\right)^{(2/3)}. \tag{52}$$

Table 4 shows that the true spectrum [15] matches well with the conventional WKB results for $n \geq 5$[2] Such a mismatch at low $n$ implies that the naive analogy of EQC between $|x|$ and QHO is not correct.

Clearly, this exercise indicates that we must introduce some modifications on the quantum periods to incorporate the derivative singularity at $x = 0$. In fact, our modified proposal for the hydrogen atom pseudo-momentum indirectly implies such a modification by treating such potentials on a half real line $\mathbb{R}^+$.

---

[1]We thank Katsushi Ito for clarifying this point.

[2]This is consistent with the WKB approximation, a semi-classical approximation, which is expected to match well for higher quantum numbers.

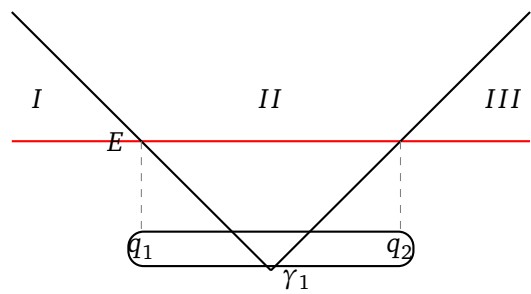

Figure 2: WKB loops for potential $V = 3|x|$.

Interestingly, the TBA equations for potentials with single and double poles on $\mathbb{R}^+$ is discussed in Ref. [13]. We will see that these potentials will capture the features of $|x|$ potential for a suitable choice of the parameters.

### 4.3  TBA equation for a potential with Single and Double Pole

We will briefly review the Schrödinger type equation with polynomial potentials with simple pole and a centrifugal term [13]:

$$\left(-\hbar^2\frac{d^2}{dx^2} + x^{s+1} + \sum_{a=1}^{s+2} u_a x^{s+1-a} + \hbar^2\frac{l(l+1)}{x^2}\right)\psi(x) = 0. \tag{53}$$

Here $x \geq 0, s \geq 0$, $l$ is any real number and $u_a$'s are parameters. For the following choice of parameters:

$$s = 0 \; ; \; u_1 = -E, \tag{54}$$

the eqn.(53) reduces to

$$\left(-\hbar^2\frac{d^2}{dx^2} + \frac{x^2 - Ex + u_2}{x} + \hbar^2\frac{l(l+1)}{x^2}\right)\psi(x) = 0. \tag{55}$$

Table 4: Spectrum for $|x|$ potential $n = 0$ to 9.

| $n$ | True Spectrum | Naive TBA Spectrum |
|---|---|---|
| 0 | 1.01879 | 1.1154602372253557 |
| 1 | 2.33811 | 2.320250794710102 |
| 2 | 3.2482 | 3.2616255199180713 |
| 3 | 4.08795 | 4.081810015382323 |
| 4 | 4.8201 | 4.826316143499807 |
| 5 | 5.52056 | 5.517163872783549 |
| 6 | 6.16311 | 6.167128465231806 |
| 7 | 6.78311 | 6.784454480834836 |
| 8 | 7.3721 | 7.374853108941933 |
| 9 | 7.94413 | 7.942486663292496 |

This resembles the equation for the linear potential in the limit $u_2 \to 0, l \to 0$. We will present the TBA equation for the potential (55) as discussed in [13].

Taking $x \in \mathbb{C}$ (complex domain), the equation (55) remains invariant under Symanzik rotation:

$$(x, E, u_2) \to (\omega x, \omega E, \omega^2 u_2), \quad \text{where} \quad \omega = \exp \frac{2\pi i}{s+3}\Big|_{s=0}. \tag{56}$$

From the semi classical behaviour in the limit $\hbar \to 0$, the turning points $e_1, e_2$ for

$$E = V(x) = \frac{x^2 + u_2}{x},$$

can be chosen to be in the positive real axis: $0 < e_1 < e_2$ for $u_2 \geq 0$. Note that $e_1 \to 0$ as $u_2 \to 0$. For this potential, there are two cycles as shown in Figure 3: $\gamma_1$ encircling $e_1$ and $e_2$ (classically allowed region) and $\hat{\gamma}$ encircling the pole at the origin 0 and $e_1$ (classically forbidden). The $l$ dependent centrifugal term (double pole) is responsible for a non-trivial

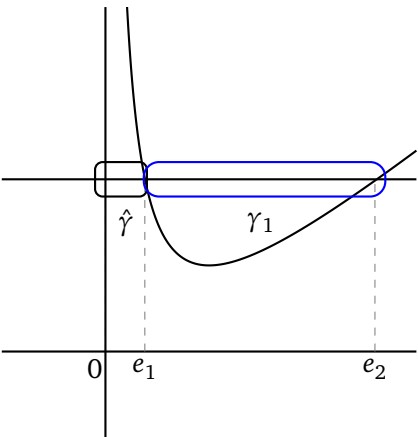

Figure 3: WKB loops for potential $V = x + u_2/x$.

monodromy of the wave function around the origin. This is very similar to our modified Bethe-like ansatz in section 2.1 for the hydrogen atom potential. In fact, the non-trivial monodromy and the Symanzik rotation symmetry leads to modification of the TBA equations[3] for the $\gamma_1$ and $\hat{\gamma}$ cycles:

$$\epsilon_1(\theta) = m_1 e^\theta - \int_{\mathbb{R}} \frac{\log(1 + \omega^{3/2} e^{2\pi i l} e^{-\hat{\epsilon}(\theta')}) + \log(1 + \omega^{3/2} e^{-2\pi i l} e^{-\hat{\epsilon}(\theta')})}{\cosh(\theta - \theta')} \frac{d\theta'}{2\pi},$$

$$\hat{\epsilon}(\theta) = \hat{m} e^\theta - \int_{\mathbb{R}} \frac{\log(1 + e^{-\epsilon_1(\theta')})}{\cosh(\theta - \theta')} \frac{d\theta'}{2\pi}. \tag{57}$$

Here $m_1, \hat{m}$ are the $0^{th}$ order WKB periods chosen with orientation so that they are both real and positive:

$$m_1 = \frac{1}{i} \oint_{\gamma_1} \sqrt{\frac{x^2 - Ex + u_2}{x}} dx \; ; \; \hat{m} = \oint_{\hat{\gamma}} \sqrt{\frac{x^2 - Ex + u_2}{x}} dx. \tag{58}$$

As discussed in section 4, the classically allowed period $\Pi_{\gamma_1}$ is not Borel summable along the positive real axis of $\hbar$. So, the period is resummed by taking average of two Borel resummations(lateral and median resummation(39)) calculated just after and before crossing the

---

[3]Interested readers can see [13] for details

discontinuity along the positive real $\hbar$ axis. For the above TBA equations, the median resummation is [13]

$$\frac{1}{\hbar}\mathcal{B}_{med}(\Pi_{\gamma_1})(\hbar) = m_1 e^\theta + P \int_{\mathbb{R}} \frac{\log(1 + \omega^{3/2} e^{2\pi i l} e^{-\hat{\epsilon}(\theta')})(1 + \omega^{3/2} e^{-2\pi i l} e^{-\hat{\epsilon}(\theta')})}{\sinh(\theta - \theta')} \frac{d\theta'}{2\pi}, \quad (59)$$

where P is the principal value of the integral which can be computed using the formula:

$$P \int_{\mathbb{R}} \frac{f(\theta')}{\sinh(\theta - \theta')} d\theta' = \lim_{\delta \to 0} \int_{\mathbb{R}} \frac{\sinh(\theta - \theta')\cos(\delta)}{\sinh^2(\theta - \theta')\cos^2(\delta) + \cosh^2(\theta - \theta')\sin^2(\delta)} f(\theta') d\theta'. \quad (60)$$

Note that the TBA equations are same for any integer $l$. We will now present the validation of our numerical computation of the Borel resummed periods and our results on the quantum periods for $V(x \geq 0) \propto x$ (linear potential).

### 4.3.1 Numerical Computation of quantum periods

To solve the TBA system, we used the Gaussian Interpolation technique presented in Appendix B of [11] with $2^{12}$ points randomly distributed around $\theta = 0$ instead of the Fourier transform method used in [13].

First, we validated our numerical code on quantum periods by confirming that the Voros spectrum for $u_1 = -3, u_2 = 1, l = -2/5$ matched with the Table 2 in [13] obtained using Bohr-Sommerfeld quantization:

$$\frac{1}{\hbar}\mathcal{B}_{med}(\Pi_{\gamma_1})(\hbar) \sim 2\pi(k + 1/2), \qquad k \in \mathbb{Z}_{\geq 0}. \quad (61)$$

Using our validated numerical code, we can compute the quantum Borel resummed periods for other values of the parameters: $u_1, l$. In order to to obtain the Voros spectrum, Bohr-Sommerfeld quantization(61) is not correct. It is not obvious how to generalise the Voros-Silverstone connection formulae and address the exact quantization condition (EQC)for such potentials with regular singularity at the origin. This is where the knowledge of the quantum periods for the linear potential $V(x \geq 0) \propto x$ with known true spectrum will be useful. The details of our proposed EQC for such singular potentials and the validation with $|x|$ spectrum will be presented in the following section.

As the EQC validation will require the quantum periods for the $|x|$ potential, we will need the numerical plots of quantum periods when $l \to 0, u_2 \to 0$. However, this limit runs into several singularities which need regularisation. Some of these details are discussed in Appendix A for completeness. We circumvent such singularities by keeping $u_2, l$ small but finite in our computation. The numerical plots for the quantum periods for $E = 1, u_2 = 10^{-8}, l = 10^{-5}$ are in Figure 4. A mathematica file containing this computation is linked on the arXiv page as an ancillary file. We will now address the EQC for potentials with singular behaviour near the origin.

## 5 Exact quantization condition for potential with singularities

Even though, we obtained the quantum periods $\Pi_{\gamma_1}, \Pi_{\hat{\gamma}}$ solving numerically the TBA system of equations(57), we have no idea how to write Voros-Silverstone connection formula to deduce the exact quantization condition (EQC) for the potentials with single and double pole at the origin.

To deal with the pole at origin, our modified Bethe-like ansatz proposal for hydrogen atom in section 2 gives a clue. There, the pseudo-momentum(24) with an additional orbital angular

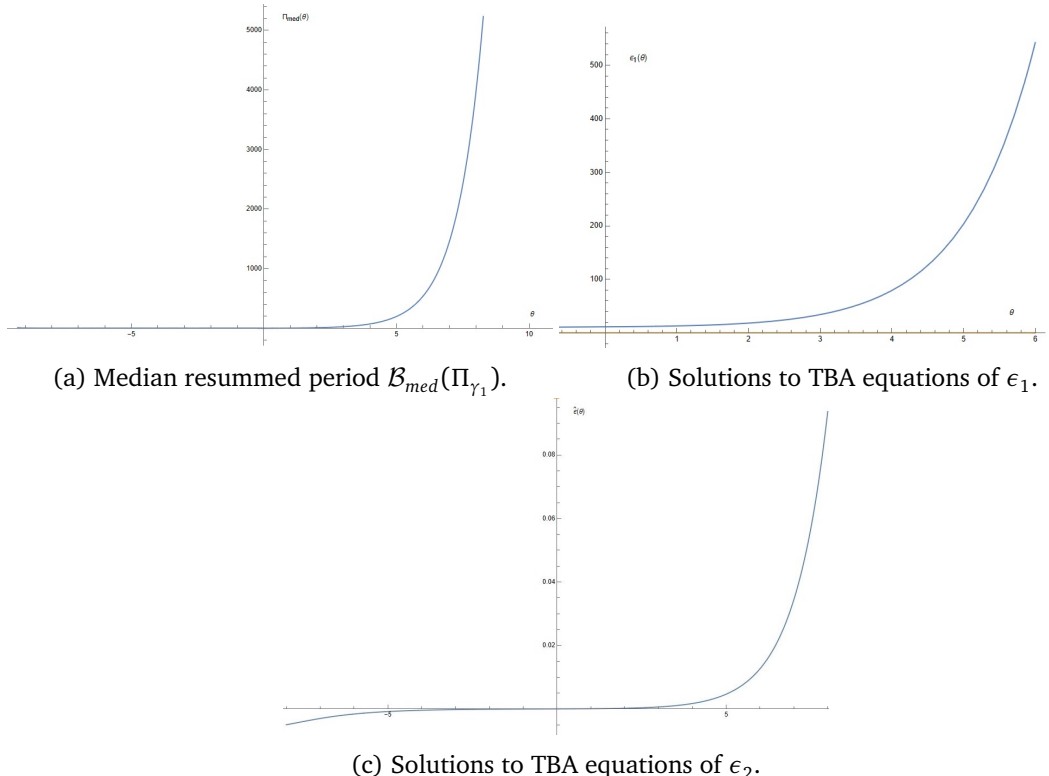

(a) Median resummed period $\mathcal{B}_{med}(\Pi_{\gamma_1})$.

(b) Solutions to TBA equations of $\epsilon_1$.

(c) Solutions to TBA equations of $\epsilon_2$.

Figure 4: $V(x) \sim |x|$ when $E = 1, u_2 = 10^{-8}, l = 10^{-5}$.

momentum dependent simple pole at the origin reflects the zero of the wave function at the origin. This suggests that the EQC must contain additional correction to the quantum periods enclosing the origin. *This is not needed for polynomial potential which has no singular behaviour at the origin.*

From our modified Bethe-like ansatz(16) as well as the exact solution for the symmetric potential $V(x) = -\frac{1}{|x|}$ [20, 21], the quantization condition for $l = 0$ is

$$\oint p \, dx = 2\pi\hbar(N+1), \qquad N = 0, 1 \dots . \tag{62}$$

The additional $2\pi\hbar$ in the above condition leads us to putforth the following proposal.

**Proposal 3:** The correction to the quantum period for the symmetric potential $V(x) = \frac{1}{|x|} + \frac{l(l+1)}{x^2}$ due to singular behaviour at the origin for $l = 0$ is

$$\Pi_0 = 2\pi\hbar. \tag{63}$$

Technically this should also be the correction for $l = -1$ as the potential is unchanged.

In order to work out EQC, we would like to go to domain $x \in (-\infty, \infty) \in \mathbb{R}$ where the wave function decays as $x \to \pm\infty$. Clearly, the TBA system on $\mathbb{R}^+$ is symmetrically mirrored about the origin. There are two more loops in $\mathbb{R}$ domain, as shown in Figure 5 which we will denote by $\gamma_{1-}, \hat{\gamma}_-$. From the symmetry $V(x) = V(-x)$, we expect

$$\Pi_{\gamma_1} = \Pi_{\gamma_{1-}}, \quad \Pi_{\hat{\gamma}} = \Pi_{\hat{\gamma}_-}.$$

However, the TBA system should continue to be exactly the same for $\Pi_{\gamma_1}, \Pi_{\hat{\gamma}}$ and their negative analogue. In fact, this equivalence is mainly due to the fact that the *origin is not a turning point* and hence should not contribute any additional Borel resummation discontinuities in the periods containing it. We see that from the period structure, the TBA system for

$V(x \geq 0) = x + 1/x$ is analogous to that of the TBA system for the symmetric double well potential (see Figure 5), except for the singular behaviour at the origin. Hence we propose the following for the classically forbidden cycle $\hat{\Gamma}$ between the two turning points $\pm e_1$:

**Proposal 4:**

The quantum period for the classically forbidden cycle $\hat{\Gamma}$ between the turning points $\pm e_1$ will be

$$\Pi_{\hat{\Gamma}} = \Pi_{\hat{\gamma}} + \Pi_{\hat{\gamma}_-} + i\Pi_0 \,, \tag{64}$$

with the same $\Pi_0$(63) correction for $l = 0$.

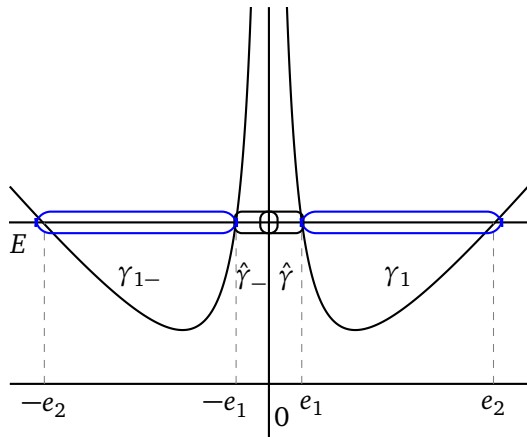

Figure 5: WKB loops for potential $V = |x| + \frac{1}{|x|}$.

As the potential in eqn.(55) near the origin resembles the symmetric hydrogen atom, we are justified to add the same $\Pi_0$(63).

Thus there are 3 nontrivial periods $\Pi_{\gamma_1}, \Pi_{\gamma_{1-}}, \Pi_{\hat{\Gamma}}$ for the potential in Figure 5. The period structure resembles the symmetric quartic oscillator in the minimal chamber. So, the same Zinn-Justin's EQC derived in [11] will be applicable:

$$\cos\left(\mathcal{B}_{med}(\Pi_{\gamma_1})/\hbar\right) \pm \frac{1}{\sqrt{1 + \exp[-\frac{i}{\hbar}\Pi_{\hat{\Gamma}}]}} = 0 \,. \tag{65}$$

Substituting for $\Pi_{\hat{\Gamma}}$(64) with $\Pi_0$(63) we get:

$$\cos\left(\mathcal{B}_{med}(\Pi_{\gamma_1})/\hbar\right) \pm \frac{1}{\sqrt{1 + \exp[-\frac{i}{\hbar}2\Pi_{\hat{\gamma}} + 2\pi]}} = 0 \,. \tag{66}$$

Now we need to fix $\pm$ sign in the EQC.

Recall for the polynomial potentials, which are regular at the origin, $\mathcal{B}_{med}(\Pi_{\gamma_1})$ obeys QHO quantization condition when $\exp[-\frac{i}{\hbar}\Pi_{\hat{\Gamma}}] \rightarrow 0$. This fixes the sign as $+$ in the EQC (66) for quartic polynomial potential [11].

However in our case with the singularity at the origin, $\mathcal{B}_{med}(\Pi_{\gamma_1})$ obeys quantization condition(62) when $\exp[-\frac{i}{\hbar}\Pi_{\hat{\Gamma}}] \rightarrow 0$. This requires the minus sign in (66). Hence the EQC to determine Voros spectrum for potential (55) is

$$\cos\left(\mathcal{B}_{med}(\Pi_{\gamma_1})/\hbar\right) - \frac{1}{\sqrt{1 + \exp[-\frac{i}{\hbar}2\Pi_{\hat{\gamma}} + 2\pi]}} = 0 \,. \tag{67}$$

Solving the above EQC for $E = 1, u_2 = 10^{-8}, l = 10^{-5}$ gives the Vorus spectrum. For this choice of parameters, $\Pi_{\hat{\gamma}} \sim \mathcal{O}(10^{(-4)})$ (as seen in Figure 4c). Hence we have neglected them in determining the Voros spectrum. Our numerical computations is tabulated in Table 5 [4] and

---

[4]Mathematica code of this computation can be found on the arXiv page as an ancillary file.

Table 5: The numerically computed Voros spectrum for the $|x|$ potential with $l = 10^{-5}, u_2 = 10^{-8}$ compared to the true spectrum.

| $n$ | Computed $\theta_n$ | True $\theta_n$ |
|---|---|---|
| 0 | 0.02852 | 0.02792 |
| 1 | 1.26107 | 1.27401 |
| 2 | 1.76443 | 1.76715 |
| 3 | 2.11220 | 2.11207 |
| 4 | 2.35925 | 2.35919 |
| 5 | 2.56402 | 2.56272 |
| 6 | 2.72669 | 2.72787 |
| 7 | 2.87390 | 2.87165 |

they match very well with the true spectrum of $|x|$ [14, 15]. Clearly, this validation suggests that our proposed EQC is applicable for general potentials $V(x \geq 0) = x + u_2/x$. Although we focused on the limit $l \to 0$ to reproduce spectrum for the $|x|$ potential, we could determine Voros spectrum for general potentials $V(x \geq 0) = x + u_2/x + \hbar^2 l(l+1)/x^2$ with non-zero centrifugal for $l > 0$. It appears from our modified Bethe-like ansatz(16), that the correction to the *proposal 3*(63) for $l > 0$ is

$$\Pi_0^{(l)} = 2\pi(l+1)\hbar.$$

Hence, using our numerical code, we can obtain Voros spectrum by including the above correction to $\Pi_{\hat{\Gamma}}$(64) in the proposed EQC(67). This elaborate exercise shows that the EQC can be constructed for the general potential $V(x \geq 0)$ in the Schrodinger equation (53). In fact, we can choose the parameters(53) so that the zeros and the turning points on the positive real line (minimal chamber). Similar to what we did for $|x| + 1/|x|$, we go back to domain $x \in (-\infty, \infty)$ to draw a symmetric potential with $2s + 3$ cycles. For all these potentials, we need to modify the EQC of the smooth polynomial potential of degree $2s + 4$. Near the origin, it is only the simple pole and centrifugal term which will contribute. Hence, $\Pi_{\hat{\Gamma}}$ near the origin must include $\Pi_0^{(l)}$ correction in the EQC. For highly excited states ($\theta \to \infty$), the EQC should reduce to the Bohr quantization condition(62) applicable to the singular potentials.

Even though the methodology is straightforward, the computation of quantum periods and Voros spectrum for higher degree polynomials gets tedious.

## 6   Conclusion

In this article, first, we reviewed Bethe-like approach for quantum harmonic oscillator (QHO) and then proposed a modification for the hydrogen atom pseudo-momentum(24). This neatly reproduced the energy spectrum and the wave functions. However, for the higher degree polynomial potentials Bethe-like ansatz fails.

We briefly reviewed WKB method, 'Thermodynamic Bethe ansatz' (TBA) method along with exact quantization conditions (EQC) leading to the spectral solutions for smooth polynomial potentials. Even though the generalisation of TBA equations for the potentials with simple and double poles [13] is known, the EQC has not been derived.

We took the symmetric form of the potential $V(x \geq 0)$(53) in the domain $x \in (-\infty, \infty)$. Such a potential will have $2s + 3$ cycles including a cycle $\hat{\Gamma}$ enclosing the origin. In this article, we focused on the potential for $s = 0$.

Taking hints from our proposed Bethe-like ansatz for the hydrogen atom, we put forth proposal 4 in section (5) stating additional correction to the quantum period $\Pi_{\hat{\Gamma}}$(64). Further, we

modified the existing quartic polynomial potential EQC by imposing Bohr-Sommerfeld quantization condition applicable for singular potentials at the origin. Our proposed EQC(67), for the potentials with single and double pole was validated by explicitly matching with the true spectrum for the $|x|$ potential corresponding to the appropriate choices of the parameters (see Table. 5). Thus we have validated our EQC for the potential (53) when $s = 0$.

Even though we elaborated for the $s = 0$ potentials, our arguments should be generalisable for the potentials with $s > 0$ as well. This requires computation of the quantum periods [13] and the modification of the smooth polynomial potential (of degree $2s + 4$) EQC [11]. The numerical computations do get cumbersome and we will take it up in future. Such an exercise could help us to validate the $|x|^3$ and other odd power Voros spectrum obtained using spectral determinant approach [5].

# Note added

After uploading this paper on arxiv, a recent paper [22] was kindly brought to our notice by one of the authors of that paper. Here they have derived a EQC for potential with regular singularity, although from a different Wronskian based approach.

# Acknowledgements

We would like to thank Katsushi Ito for discussions on TBA propagator. We are grateful to Marcos Mariño for sharing his notes which turned out very valuable. We would also like to thank the referees for their valuable comments.

**Funding information** PR would like to acknowledge the ICTP's Associate programme where significant progress on this work happened during her visit as senior associate. AA would like to acknowledge IIT Bombay for supporting travel to *Integrability in Gauge and String Theory 2022* conference where parts of this work were presented. PM is supported by a scholarship from the Inlaks Shivdasani Foundation.

# A Regularisation of TBA for $|x|$

In section 4.2, we saw the TBA equation(57) for a linear potential with a single and double pole :

$$V(r) = r - E + \frac{u_2}{r} + \frac{\hbar l(l+1)}{r^2},\qquad(68)$$

which takes the form:

$$\epsilon_1(\theta) = m_1 e^\theta - \frac{1}{2\pi}\int_{\mathbb{R}}\frac{\log(1 + e^{-2\hat{\epsilon}(\theta')} - 2\cos(2\pi l)e^{-\hat{\epsilon}(\theta')})}{\cosh(\theta - \theta')},$$

$$\hat{\epsilon}(\theta) = \hat{m}e^\theta - \frac{1}{2\pi}\int_{\mathbb{R}}\frac{\log(1 + e^{-\epsilon_1(\theta')})}{\cosh(\theta - \theta')}.\qquad(69)$$

In order to reproduce the Hamiltonian for the pure $|x|$ potential, we would like to take $u_2 = 0, l = 0$ or $-1$. However, this leads to highly singular behaivour and hence we need

to carefully take the limit $u_2 \to 0$ and $l \to 0$ or $l \to -1$ where the single and double pole seemingly vanish. In this limit,

$$m_1 \to \frac{4}{3}E^{3/2}, \qquad \hat{m} \to 0.$$

With $E = 1$, (69) reduces to

$$\epsilon_1(\theta) = \frac{4}{3}e^\theta - \frac{1}{2\pi}\int_{\mathbb{R}} \frac{\log(1 + e^{-2\hat{\epsilon}(\theta')} - 2e^{-\hat{\epsilon}(\theta')})}{\cosh(\theta - \theta')}d\theta'$$

$$\hat{\epsilon}(\theta) = -\frac{1}{2\pi}\int_{\mathbb{R}} \frac{\log(1 + e^{-\epsilon_1(\theta')})}{\cosh(\theta - \theta')}d\theta'.$$

However, this TBA system is highly singular: as $\theta \to \infty, \hat{\epsilon} \to 0$ and $\log[(1 - e^{-\hat{\epsilon}(\theta')})^2] \to -\infty$. Thus additional regularisation is needed before this TBA system can be used. Following [16, 19] the regularisation of the singular term $\epsilon_1$ must be done by subtracting a factor of $\log(2\pi l)$. We see this as follows :

$$\epsilon_1(\theta) \sim -\log(2\pi l e^{-A(\theta)} + \mathcal{O}(l^2)),$$
$$\hat{\epsilon}(\theta) \sim -\log(1 + 2\pi l B(\theta) + \mathcal{O}(l^2)). \tag{70}$$

Then expanding the TBA equations in $l$, we have

$$A(\theta) - \log(2\pi l) = \frac{4}{3}e^\theta - \frac{1}{2\pi}\int_{\mathbb{R}} \frac{d\theta'}{\cosh(\theta - \theta')}\log(1 + B^2(\theta')) - \log(2\pi l)$$

$$B(\theta) = \frac{1}{2\pi}\int_{\mathbb{R}} \frac{d\theta'}{\cosh(\theta - \theta')}e^{-A(\theta')}. \tag{71}$$

Thus we see that the divergent term in $\epsilon_1(\theta)$ ends up cancelling on both sides, leaving us with a system of equations that is no longer divergent. This regularisation, although initially appearing to be valid only for the even states with $l = 0$ carries through exactly in the same way and gives us the same equations if instead we choose to expand around $l = -1$. This is due to the periodicity of the only explicit $l$ dependence in the TBA equations is given by the $\cos(2\pi l)$ term.

Further, this is the same TBA system shown in [19] (up to a overall constant shift of $\theta$), which was shown to be solved by the Airy functions :

$$e^{-A(\theta)} = -2\pi \frac{d}{dz}Ai^2(z), \tag{72}$$

$$B(\theta) = -2\pi \frac{d}{dz}Ai(e^{i\pi/3}z)Ai(e^{-i\pi/3}z), \tag{73}$$

with $z = e^{\frac{2}{3}\theta}$. It was argued that $e^{-A(\theta)}$ must be the correct spectral determinant for the problem, since it vanishes at those values of $\theta$ where $Ai(z) = 0$ or $Ai'(z) = 0$ which correspond to the true spectrum. To get a direct derivation of this from the TBA system, let us examine what happens to the Bohr Sommerfeld quantisation under the regularisation scheme. The shift from $\epsilon_1$(69) to $\Pi_{\gamma_1}$ involves a rotation $\theta \to \tilde{\theta} = \theta + i\pi/2$. This takes the form [13]

$$\frac{1}{\hbar}\mathcal{B}_{med}(\Pi_{\gamma_1}) = \frac{4}{3}e^{\tilde{\theta}} + \frac{1}{2\pi}P\int_{\mathbb{R}} \frac{d\theta'}{\sinh(\tilde{\theta} - \theta')}\log(1 + e^{-2\hat{\epsilon}(\theta')} - 2\cos(2\pi l)e^{-\hat{\epsilon}(\theta')}),$$

$$\frac{1}{\hbar}\mathcal{B}_{med}(\Pi_{\gamma_1}) = 2\pi(n + 1/2), \quad n = 0, 1, 2\ldots$$

However under the regularisation scheme(70), we have

$$\mathcal{B}_{med}(\Pi_{\gamma_1}) = \mathcal{B}_{med}(A(\theta)) + \log(2\pi l) = 2\pi\hbar(n + 1/2). \tag{74}$$

This implies that the points in the spectrum $\{\theta_i\}$ which solve the Bohr Sommerfeld condition satisfy

$$e^{-\mathcal{B}_{med}(A(\theta))} = \mathcal{O}(2\pi l), \tag{75}$$

which must vanish when $l \to 0$. Hence, $e^{-\mathcal{B}_{med}(A(\theta))}$ must be the spectral determinant for the problem.

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
