# Peer review of "Spectral solutions for the Schrödinger equation with a regular singularity"

_SciPost Physics Core, doi:SciPost Phys. Core 7, 041 (2024)_

## Round 1 · Referee Report · Anonymous (Referee 1) · 2024-4-8

Strengths

  1. The paper is well written and self explanatory with sufficient technical details.

  2. It contains useful pedagogical reviews of Bethe-like Ansatz and WKB method.

  3. The paper addresses an important question on Exact Quantisation Condition in the presence of singular potential. A novel proposal for modification of Bethe-like Ansatz for obtaining spectrum of Hydrogen atom is presented in the paper.

Weaknesses

  1. Few equations in the paper contain minor typographical errors.

Report

In this paper the authors have studied spectral solution for one-dimensional Schrödinger equation with potentials which have regular singularity at the origin. They have proposed a modification for Bethe-like Ansatz applicable to Hydrogen atom which has a potential of the form $\frac{1}{r}$. The authors have also presented Exact Quantisation Condition for potentials with single and double poles at the origin. The results obtained in this paper are potentially useful in the studies of integrable models. I recommend this paper to be published in SciPost Physics Core. I would like to point out some probable typos in some places, which the authors should verify and rectify them before publication:

  1. In the equation below Eq.(2.4) there will be a factor of $\frac{i}{\hbar}$ in side the exponential and the full equation should be $\psi(z)\propto\exp\left(\frac{i}{\hbar}\int p(z)\mathrm{d}z\right) = \exp\left(-\frac {i}{\hbar}\frac{\left(z-a\right)^{-n+1}}{n-1}\right)$.

  2. Similar $\frac{i}{\hbar}$ factor should be present in the first equality of Eq.(2.13).

  3. Numerator of the first term in Eq.(2.14) will not have $r$.

  4. To be consistent with Eq.(2.14), $b$ in Eq.(2.16) should be $b= -\frac{e}{4\pi\epsilon_{0}}$. The authors are requested to check the sign of $b$ given below Eq.(2.16).

  5. The index in the summation symbol appearing in Eq.(3.3) should be $i$ instead of $j$.

  6. I am confused with the upper limit of the integration appearing in Eq.(3.10). $\phi_{\infty}$ needs to be defined.

  7. In Eq.(4.5) the subscript on the left hand side of the equality will be $a$ instead of $i$.

  8. There will be no $\hbar$ in the denominator in the last equality of Eq.(4.10).

In addition, the authors may choose to address the following queries if they wish:

  1. While taking the classical limit in Eq.(2.2), the term $i\hbar p'$ is dropped. However from Eq.(2.3) we see that expression of $p(x)$ contains $\hbar$. It will be better if some clarification regarding scalings of $\psi(x)$, $E$ and $V(x)$ appearing in Eq. (2.1) with respect to $\hbar$ is provided.

  2. In the naive TBA approach for $|x|$ potential, as the authors have shown, the spectrum matches well with the expected answer for $n\ge 5$. Is there any reason why the mismatch is more for lower values of $n$ only and the analogy of EQC between $|x|$ and harmonic oscillator holds in the large $n$ regime?

---

## Round 1 · Referee Report · Anonymous (Referee 2) · 2024-4-24

Strengths

This manuscript provides a modified Bethe like ansatz to study quantum mechanical problems with regular singularity and provides an exact quantisation condition.

Weaknesses

The manuscript can be trimmed to focus on the problem.

Report

Authors provide an exact quantisation condition for quantum mechanical problems with regular singularity at origin. While the procedure uses the Thermodynamic Bethe ansatz(TBA) technique, the manuscript contains a section on the WKB and exact WKB method which is not quite directly relevant except for the Voros symbols that appear in the TBA. The details of (exact) WKB has appeared in many articles and I believe the section can be pruned and relevant formulae can be added in the TBA section.

I recommend publication of shortened version in SciPost Physics Core

Requested changes

  1. In section 2 when authors analytically continue $p(x)$ to $p(z)$, they should do the same with $\psi(x)$ to $\psi(z)$ and $V(x)$ to $V(z)$. Similarly, the table 1 should have $z$ and not $x$. Similarly, in eq.(2.9) the residue should be at $z_j$ and not $x_j$.
  2. $V_eff$ in 2.13 has a spurious $r$ in the numerator.
  3. The sum is over different variable in eq.(3.3).
  4. In eq.(3.4) and (3.5), $\Pi_{\gamma,n}$ cannot be a function of $\hbar$.

Recommendation

Ask for minor revision

---

## Round 2 · Referee Report · Anonymous (Referee 2) · 2024-5-19

Report

I recommend publication of the revised manuscript in SciPost Core.

Recommendation

Publish (meets expectations and criteria for this Journal)

---

## Round 2 · Referee Report · Anonymous (Referee 1) · 2024-5-19

Report

I am satisfied with the modifications made by the authors. I recommend the manuscript be published in SciPost Physics Core.

Recommendation

Publish (easily meets expectations and criteria for this Journal; among top 50%)

---

## Round 2 · Author Response

We thank both the referees for their valuable comments.

We have revised the manuscript incorporating the minor suggestions and queries put forth by referee 1 and referee 2. The places where we have made the corrections are indicated in blue color for clarity in the .pdf format of the revised manuscript. Also we have pruned section 3 removing `conventional WKB' subsection as suggested by referee 2.

We hope the revised paper can be accepted for publication in SciPost Physics Core.

---

## Round 2 · List of Changes

We have addressed the typo in equation (2.14).

Referee 2 correctly pointed out that $\Pi_{\gamma,n}$ cannot be a function of $\hbar$. We have rectified this mistake accordingly.

Following the observation made by Referee 1, we have corrected the typos in equations (2.5, 2.10,2.14,2.15, 2.30, 2.31, 4.5,4.6 4.10) and the $b$ value in the line after eqn.(2.17)

We have corrected the definition of the parameter \(b\) before eqn.(2.19) to be consistent.

We have cited refs.[6,8] at the end of second paragraph in section 3.1.

We have added a line explaining the limits of integration for the Borel resummation in eqn. 3.5.

We have added a line in the `Acknowledgements' to thank both the referees.

In addition to these corrections, we have substantially reduced the discussion in section 3 on the conventional and exact WKB methods. Instead of discussing many details that are well known and peripheral to the paper as pointed out by Referee 2, we mainly discuss the definitions of the quantum periods and their Borel resummation in order to be self contained.

We have removed Fig. 1 in the previous manuscript related to the WKB theory of the quantum harmonic oscillator also.

We have also changed \(x \to z\) in section 2 wherever required.

We have added footnote 2 in page 15 to address one of the queries of Referee 1.

---

## Editorial Decision

published